# Spatio-Temporal Analysis of Forest Growing Stock Volume and Carbon Stocks: A Case Study of Kandry-Kul Natural Park, Russia

**Aleksandr Volkov, Larisa Belan, Ekaterina Bogdan \*, Azamat Suleymanov, Iren Tuktarova, Ruslan Shagaliev and Dilara Muftakhina**

Ufa Decarbonisation Technology Center of State Petroleum Technological University, Kosmonavtov str., 1, Ufa 450064, Russia; volkovufa@mail.ru (A.V.); belan77767@mail.ru (L.B.); filpip@yandex.ru (A.S.); umrko@mail.ru (I.T.); shagaliev@rambler.ru (R.S.); muftakhina.d@gmail.com (D.M.)

\* Correspondence: eavolkova@bk.ru

**Abstract:** This paper presents the evaluation and digital mapping of forest growing stock volume (GSV) and carbon stocks of the natural park Kandry-Kul (Republic of Bashkortostan, Russia). The field observations were conducted in the process of forest surveys in 1994 and 2018. According to these materials, we identified the predominant tree species in each studied plot. Then, we produced the digital maps of GSV and carbon stocks for each study year and calculated the annual increment. The results showed that birch (*Betula pendula*) and pine (*Pinus sylvestris*) were the dominant species in the studied park. The comparison of the two maps showed that the maximum annual GSV between 1994 and 2018 occurred in areas with a fairly small GSV in 1994. These areas were characterized by young trees of different species and pines of different ages, which had not yet reached the age of maturity, when the growth of trees is significantly reduced. We found that young pine crops contributed the greatest potential of carbon sequestration, with an annual GSV of 4.2 m³/ha per year. The birch trees made a minor contribution to the sequestration potential, characterized by relatively average annual growth (1.2 m³/ha per year). The change in carbon stocks for pine stands was on average 2 t/ha per year. For birch stands, the change in carbon stocks was approximately 0.5 t/ha per year, which is 30% of the average values for the forest-steppe region. Our results provide quantitative information on the carbon accumulation rate in secondary forests under conditions of intensive recreational load.

**Keywords:** forest growing stock volume; carbon; mapping; space-time; climate change

## 1. Introduction

As a result of anthropogenic activities all over the world, there are changes in the main climatic indicators. The leading factors of climate warming are fossil fuel emissions and changes in land use [1]. Rising temperatures, changing precipitation patterns, and increased frequency of extreme weather events disrupt ecosystems by altering species distributions, agriculture, disturbing natural cycles, and increasing the risk of forest fires and pests [2]. These impacts can lead to habitat loss, biodiversity decline, reduced productivity, and jeopardized ecosystem services, highlighting the urgent need for conservation and adaptive management strategies to safeguard forest ecosystems in the face of climate change. Forests are one of the main natural stabilizing mechanisms that compensate for increased emissions of greenhouse gases into the atmosphere. Estimates of carbon stock changes in woody vegetation are important for accounting and monitoring forest ecosys-

tems, and especially for climate change mitigation [3,4]. By quantifying the carbon sequestration potential of different tree species and forest areas, research on forest properties contributes to the development of effective climate change adaptation and mitigation policies.

Cartographic support is becoming an increasingly important task in ecosystem studies, especially in a relatively rapidly changing environment. The increase in the power of electronic computers, emergence of remote sensing data, GIS technologies, and machine learning methods has greatly transformed the mapping of environmental components. Detailed surveys carried out in the last century by highly qualified specialists are valuable material for modern research and retrospective monitoring. Such archival data allow us to trace the dynamics of changes in qualitative and quantitative indicators and visualize them in a cartographic form. For example, several studies performed digital mapping the dynamics of carbon in terrestrial ecosystems using archival and modern data [5–8]. Spatio-temporal studies provide valuable insights into forest growth patterns, enabling better forest management decisions [9]. By analyzing the growth dynamics, researchers and forest managers can identify areas with high potential for sustainable timber harvesting, plan reforestation efforts, and prevent overexploitation. Additionally, studying the spatio-temporal distribution of GSV and carbon stocks helps identify areas where interventions are needed to maintain healthy forests, preserve biodiversity, and protect ecosystem services [10]. Moreover, this information is essential for accurate carbon accounting and monitoring progress towards emission reduction targets [11,12]. By analyzing historical data and trends, spatio-temporal studies enable the development of predictive models for future forest growth and carbon dynamics [13]. These models can assist in anticipating the impacts of climate change on forest ecosystems and guide adaptation strategies. They provide valuable information for policymakers, land managers, and stakeholders to make informed decisions on forest conservation, afforestation, and restoration efforts.

Landscapes of specially protected natural areas are important research sites, since they provide a regime for preserving the course of natural processes and minimal anthropogenic impacts on ecosystems. Data on the carbon sequestration potential of pristine forests provide reliable information on the spatial and temporal dynamics of primary forest productivity. Consequently, the data obtained are due to the natural course of environmental processes, primarily the response of forests to climate change.

The evaluation and mapping of forest growing stock volume (GSV) and carbon stocks support the identification of high-priority areas for conservation and restoration, facilitating targeted efforts to enhance carbon sequestration and ecosystem resilience. The overall aim of the present study was to estimate and conduct a digital mapping of GSV and carbon stocks in the natural park Kandry-Kul (Republic of Bashkortostan, Russia). The specific objectives were to: (1) determine the dominant species according to taxation plots; (2) identify the GSV and carbon stocks for 1994 and 2018 and then calculated the annual increment dynamics; (3) produce the digital maps of GSV and carbon stocks; and (4) discuss limiting factors for tree development.

## 2. Materials and Methods

### 2.1. Study Area

The research was conducted in the "Kandry-Kul Nature Park" (Tuymazinsky District, Republic of Bashkortostan, Russia) (Figure 1). The park is a popular place for recreation and the largest number of visitors to the park arrive during the hot season. As a result, the negative climatic impact in the region is exacerbated by the increased level of recreational load [14–16]. The study area is located in the west part of the republic, within the Kandrykul-Usen and Usensko-Ryasky physical-geographical districts of the Belebey upland-level typical forest-steppe district [17]. The lake Kandry-Kul is located on the territory and is the main attraction of the park. The total area of forest lands in the natural park is 1149 ha, and the area covered by forest is 940.3 ha. The forest mensuration sites are shown in Figure 1.

The territory of the natural park was developed agriculturally in the first half of the XX century, which led to the loss of meadow steppes and broad-leaved forests. Now there are small areas of oak stands left. They have been replaced by secondary birch, aspen, and linden forests. Thus, the territory of the park is an excellent example of long-term restoration of natural vegetation on abandoned arable land under conditions of active recreation.

In this entire area, timber harvesting is prohibited by the Regulations on the nature park. Cutting of single trees is allowed only upon the conclusion of specialists—forest pathologists—and in order to remove slanted trees that threaten safety. Approximately 70% of the park's forests is located in the watershed and is rarely used for recreational purposes. These forests represent a whole preserved sample of the forest characteristic of Tuimazinsky forestry—secondary birch (*Bétula*), linden (*Tília*) and pine (*Pínus*) cultures of different years of planting.

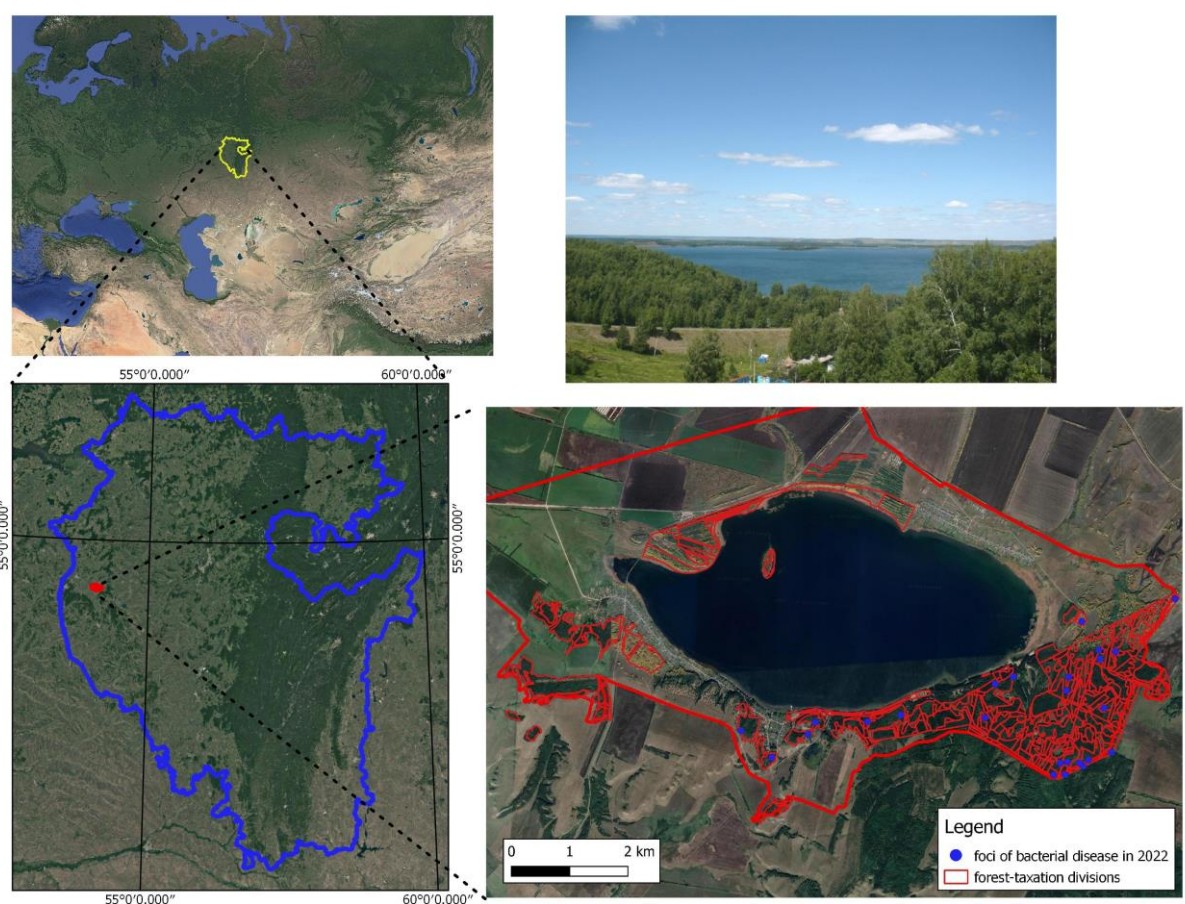

**Figure 1.** Location of the Kandry-Kul Nature Park.

The climate of the study area is characterized as moderate continental or as warm-summer humid continental (Dfb) by the Köppen climate classification [18]. Winter is cold, from mid-November. The average temperature in January is −13.8 °C. The absolute minimum is −50 °C. The average duration of steady snow cover is 134 days. The average height of snow cover is 28 cm. The average amount of rainfall during winter is 103 mm. The hottest month is July, with an average July temperature of +18.4 °C and an absolute maximum of +40 °C. The frost-free period in the city has a duration of 123 days.

### 2.2. Field Investigation and Digital Mapping

The data on GSV (m³/ha) and carbon stocks (t/ha) in each forest mensuration unit were used as the main quantitative parameter for estimation and digital mapping. The

field surveys were conducted in the process of forest surveys in 1994 and 2018 by Kandrinskoye district forestry (Tuimazinskoye forestry of the Republic of Bashkortostan). The taxation characteristics were determined according to the manual of Anuchin [19]. According to the forest taxation methodology, trees shorter than 5 m in height and with a trunk diameter of less than 5 cm were not included in this study.

An eye-measuring study of the characteristics of the wood stand was carried out. The diameter of the tree is measured using a tree caliper. The height of the tree was measured using an eclimiter. The method of determining the height of the tree is based on the fact that the ratio of the height of the tree *H*, reduced by an amount equal to the distance from the ground to the observer's eyes (*h*), to the distance from the tree to the point from which the top of the tree is viewed (*A*), is the tangent of the angle $\alpha$ formed by the horizontal position and the lines of sight (Formula (1)).

$$\frac{H-h}{A} = tg\ \alpha \tag{1}$$

To study the organic mass of forest phytocenoses, one trial plot was established in each forest area. At this site, measurements and observations were made for growth, decay, microclimate, etc., without disturbing the original condition of the ecosystem. Based on the diameter and height, we calculated the stock of raw trunk wood of the forest stand (m³/ha) by species. Then, using the percentage of participation of each tree species in the total stand stock, we obtained a formula for its composition. We calculated the annual increase in GSV in all forest plots for which data are available both in the forest management materials of the 1994 and 2018.

To obtain information about the carbon storage potential of forests, the difference between the rates of GSV for 1994 and 2018 was calculated. Next, the difference in GSV in each forest mensuration unit was divided by 24 (the number of years that passed between the two forest surveys). As a result, the annual GSV for the period 1994–2018 was obtained for each forest mensuration unit of the park. The data for anthropogenic-disturbed areas, such as areas of a children's camp and a ski complex, were excluded in the studied area. The forest plantations of these areas are significantly disturbed—partially cut down, built up with buildings and recreational facilities. The mode was used to establish the most frequent values or the middle of the interval of values, in which the largest number of values was included. The digital mapping of forest properties was performed in QGIS 3.16.1.

The carbon stocks assessment was based on 1994 and 2018 forest inventory data. Further, conversion factors (Table 1) were used to calculate carbon stocks (t/ha) based on the GSV data presented in the Methodological guidelines for quantitative determination of the amount of greenhouse gas absorption, approved by the Order of the Ministry of Natural Resources of Russia from 30 June 2017 [20]. The carbon stocks in the biomass of stands by the age groups of the dominant species was calculated by Formula (2):

$$CP_{ij} = V_{ij} \times KP_{ij} \tag{2}$$

where $CP_{ij}$ is carbon stocks in the biomass of stands of age group *i* of the dominant species *j*, t/ha; $V_{ij}$ is GSV of age group *i* of the dominant species *j*, (m³/ha⁻¹) and $KP_{ij}$ is conversion factor to calculate the carbon stocks in the biomass of stands of age group *i* of the dominant species *j*, tC/m³

**Table 1.** Conversion factors for calculating carbon stocks in tons/ha [20].

| Species | Age Group | | |
| :---: | :---: | :---: | :---: |
| | Class 1 * | Class 2 ** | Class 3 *** |
| *Pinus silvestris* | 0.435 | 0.352 | 0.329 |
| *Picea abies* | 0.614 | 0.369 | 0.351 |
| *Larix sibirica* | 0.392 | 0.371 | 0.398 |
| *Quercus robur* | 0.796 | 0.541 | 0.563 |

| | | | |
|---|---|---|---|
| *Ulmus sp.* | 0.624 | 0.477 | 0.388 |
| *Tilia cordata* | 0.624 | 0.477 | 0.388 |
| *Betula pendula* | 0.624 | 0.477 | 0.388 |
| *Populus temula* | 0.437 | 0.396 | 0.367 |

* Coniferous and hardwood—0–40 years; softwood—0–20 years. ** Coniferous and hardwood—40–60 years; softwood—20–30 years. *** Coniferous and hardwood—60–100 years; softwood—30–80 years.

Additionally, when calculating the carbon stocks, we took into account such a taxonomic indicator as bonitet, which depends directly on the productivity of the stand and is determined by the age and height.

### 3. Results

*3.1. Analysis of the Structure and Condition of Forest Stands according to Forest Inventory Materials 1994*

Figure 2 shows the spatial distribution of forest taxation divisions of the Kandry-Kul Nature Park by dominant tree species according to 1994. The results showed that birch (*Betula pendula*) and pine (*Pinus sylvestris*) were the dominant species (56.0% and 25.7%, respectively) and were evenly distributed throughout the park. In the eastern part, aspen (*Populus tremula*), alder (*Alnus glutinosa*), and other tree species dominated the habitat. Larch (*Larix sibirica*) was the predominant species and occupied the entire forest unit in the northern part.

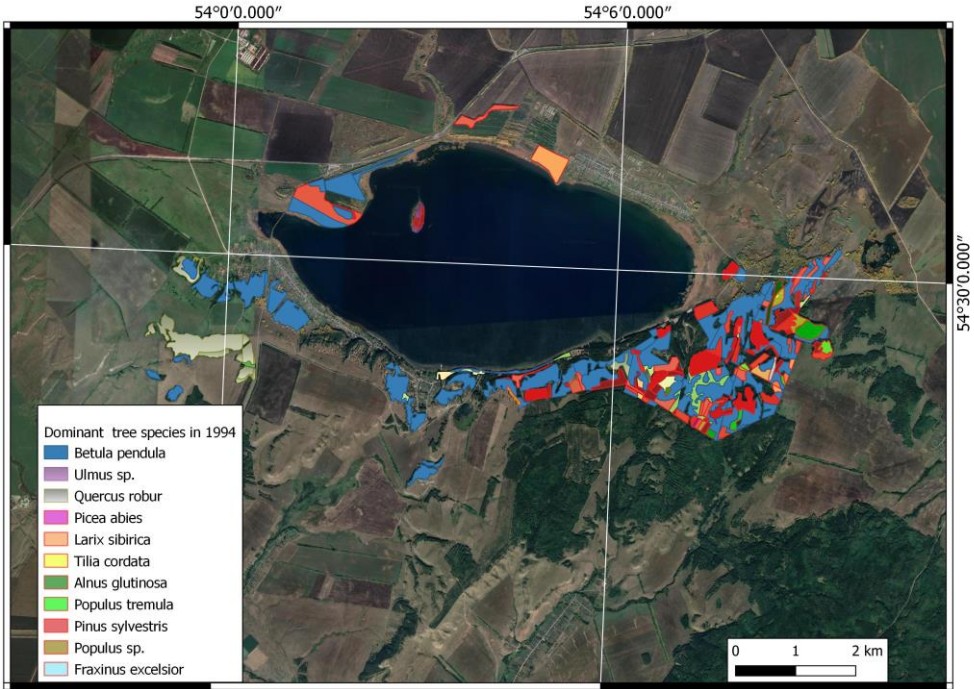

**Figure 2.** Spatial distribution of forest-taxation divisions of Kandry-Kul Nature Park by dominant tree species in 1994.

The GSV by forest taxation areas according to studies in 1994 is shown in Figure 3. For clarity, the spatial distribution of GSV is divided into 10 classes and for 1994 the step was 25 m/ha, and for 2018 the step was 34 m/ha. The largest GSV values was located in the eastern part of the territory. Here, the maximum values were concentrated in the sections under the birches (175–250 m$^3$/ha). The southern, eastern and northern parts were characterized by the lowest forest GSV (less than 100 m$^3$/ha), which was associated with young trees planted in the 1980s (sections № 68 and 67).

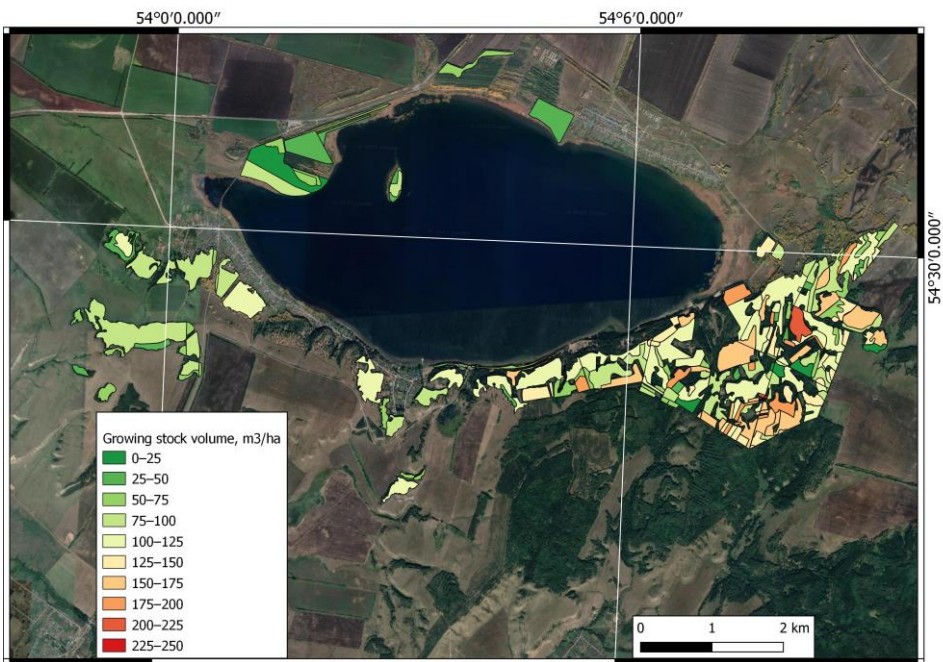

**Figure 3.** Spatial distribution of GSV (m³/ha) by forest taxation divisions of the Kandry–Kul Nature Park in 1994.

The forest GSV distribution for all tree species in 1994 is shown in Figure 4. This histogram shows that the highest GSV values ranged from 69 to 237 m³/ha, where maximum frequency of occurrence corresponded to values close to 111 m³/ha. The GSV mode value in the compartments with the predominance of birch was 133 m³/ha, while for the pine was 203 m³/ha, which was 1.5-fold higher than for the birch-dominated compartments.

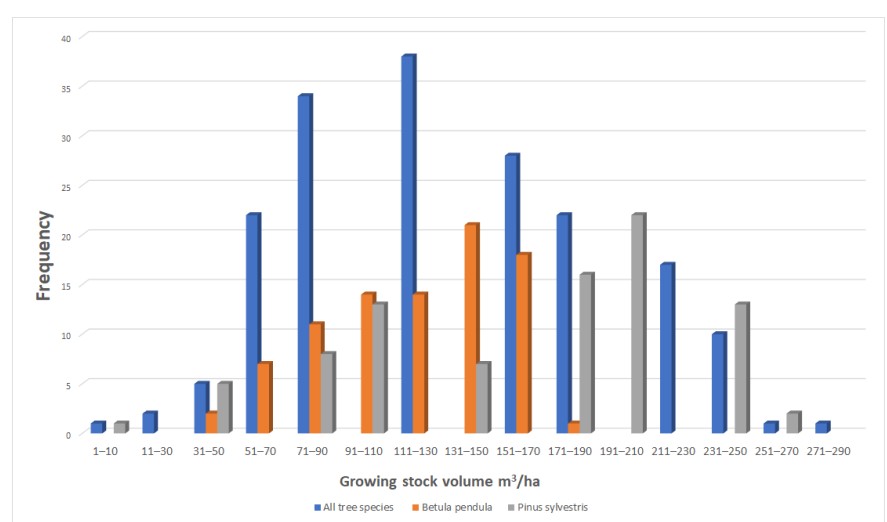

**Figure 4.** Histogram of the GSV values (m³/ha) distribution for all species, as well as by taxation divisions with a predominance of birch and pine in 1994.

### 3.2. Analysis of the Structure and Condition of Forest Stands according to Forest Inventory Materials 2018

The spatial distribution of the GSV by forest taxation divisions of the Kandry-Kul Nature Park in 2018 is shown in Figure 5. The distribution pattern of maximum values in 2018 was consistent with 1994. These sections were located on the south shore of the lake and were characterized by a predominance of pines.

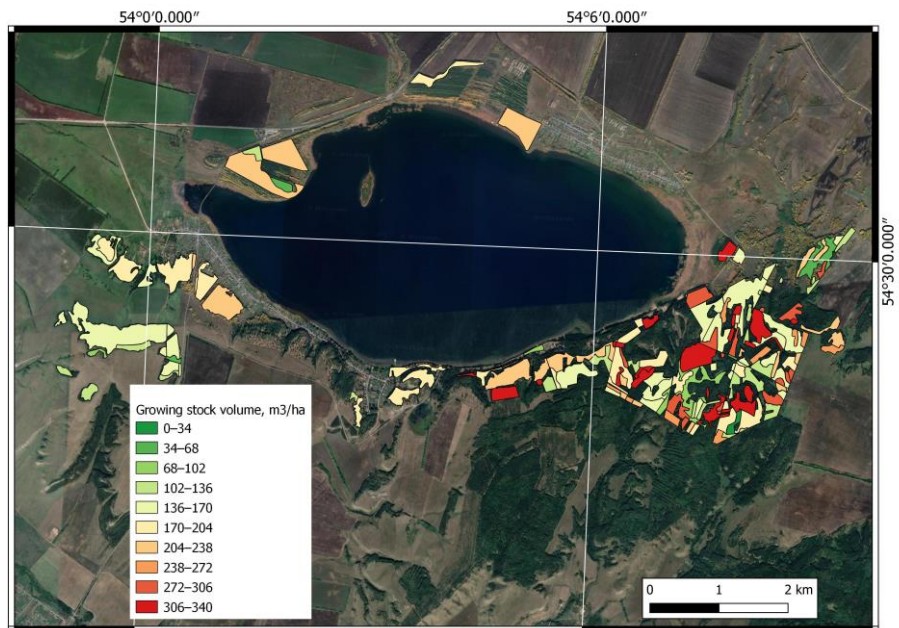

**Figure 5.** Spatial distribution of the GSV (m³/ha) by forest taxation divisions of the Kandry-Kul Nature Park in 2018.

The distribution graph of the GSV in the birch and pine-dominated sections in 2018 is shown in Figure 6. The forest GSV in the birch-dominated sites increased markedly in 2018 compared to 1994. The mode value in 2018 was 186 m³/ha, compared to 133 m³/ha in 1994, i.e., an increase of 51 m³/ha. Compared with 1994, the GSV in the compartments with the predominance of pine has also increased significantly. The distribution according to the 2018 mode value was 308 m³/ha, compared to 203 m³/ha in 1994, i.e., an increase of 105 m³/ha.

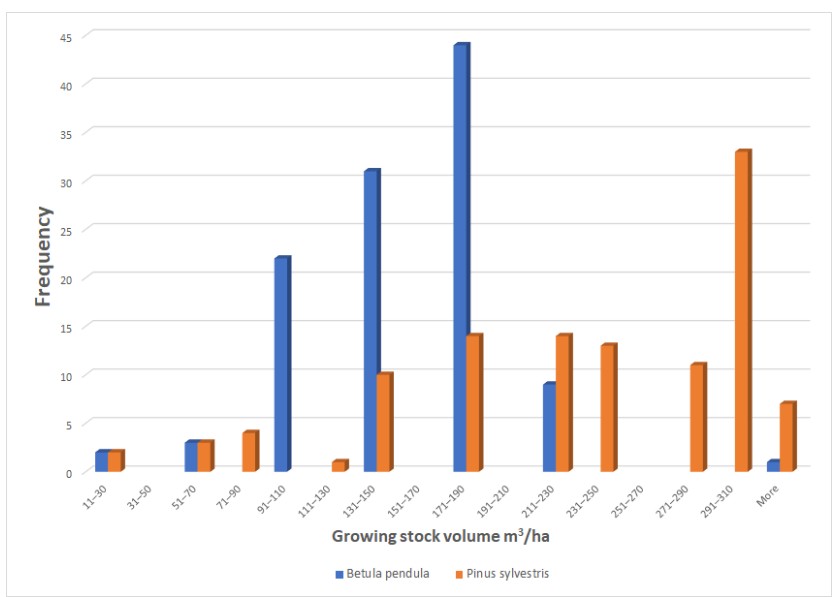

**Figure 6.** Histogram of the GSV values (m³/ha) by taxation divisions with a predominance of birch (*Betula pendula*) and pine (*Pinus sylvestris*) in 2018.

### 3.3. GSV Annual Increase in the Nature Park for the Period 1994–2018

The calculation of annual incremental growth of the GSV for the period 1994–2018, geo-referenced to the sections of 1994 are presented in Figure 7. According to the visuali-

zation, it was found that the maximum annual GSV values were 6.4–7.9 m³/ha and occurred in the stands with a fairly small GSV values in 1994, i.e., with young stands relative to the other units. These sections were characterized by the highest annual growth with young trees of different species that reappeared in the sections with previously absent plantings. The main representative of such stands were pine cultures belonging to the young trees age group, with an annual growth of GSV 4.2 m³/ha per year. The birch trees were the largest in terms of the share of area in the total area covered by forests on the territory and belonged to the mature forest stands. They had the lower value (1.2 m³/ha per year) relative to the average annual growth for the Tuimazinsky forest area (3.2 m³/ha per year), and made a minor contribution to the carbon-deposit potential.

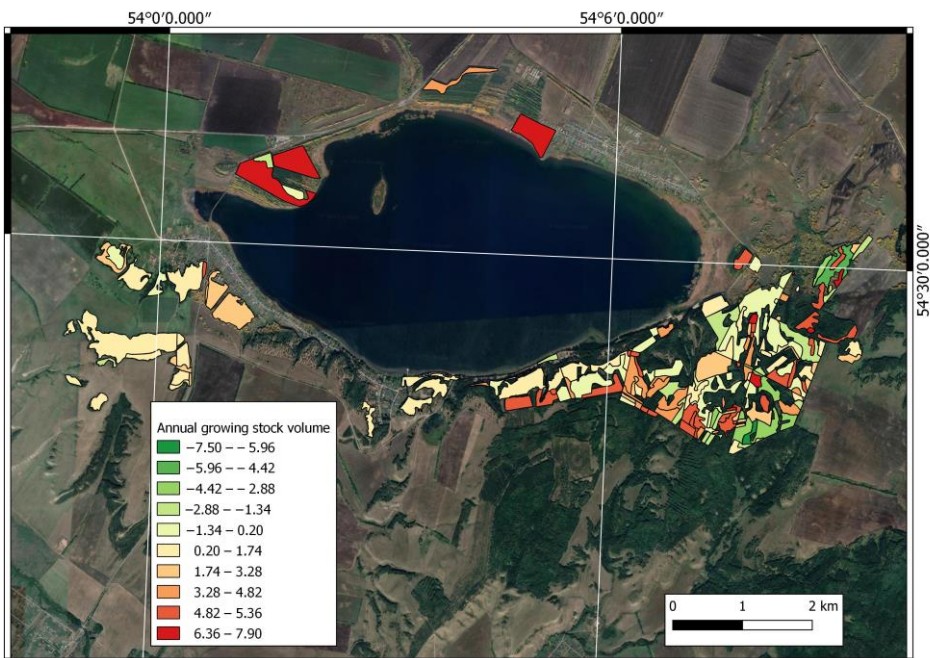

**Figure 7.** Spatial distribution of the annual GSV (m³/ha) by forest taxation divisions of the Kandry-Kul Nature Park for the period of 1994–2018.

Negative annual GSV was found in the eastern and southern parts of the territory. These forest taxonomic divisions were characterized by the predominance of birch stands. Other tree species, due to their insignificant representation in the forests, did not contribute significantly to the carbon deposition.

### 3.4. Changes in GSV and Carbon Stocks from 1994 to 2018

Table 2 shows the changes in GSV and carbon stocks values in the forest stands. According to the results, the largest carbon stocks were concentrated in pine and birch forests. Among young stands, pine stands had the highest carbon stock in 1994 (60,86 t/ha), while linden stands had the highest carbon stocks in young stands in 2018 (87,36 t/ha). Among mid-aged species, the highest carbon stock in both 1994 and 2018 was characteristic of pine trees. Additionally, pine stands had the highest carbon stocks in old stands in 2018. Clearly, birch was previously the main distributed species, but birch forests were less resistant to recreational pressures and the effects of climate change. Thus, the spread of bacterial disease resulted in the average annual change in carbon stocks for birch (*Betula pendula*) to be 4-fold lower than for pine (*Pinus sylvestris*) (Figure 8). Relative to other species there was a significant change in carbon stocks for hardwoods: linden (*Tilia cordata*) and oak (*Quercus robur*). Thus, the main tree, depositing carbon and resistant to anthropogenic pressures and climate change, were pine species. The results also show that

in birch stands the change in carbon stocks was approximately 0.5 t/ha per year, which is 30% of the average values for the forest-steppe region obtained by Shvidenko [21].

**Table 2.** Changes in GSV and carbon stock values from 1994 to 2018.

| Species | Age Group | | | | | | | | | | | |
|---|---|---|---|---|---|---|---|---|---|---|---|---|
| | Class 1 * | | | | Class 2 ** | | | | Class 3 *** | | | |
| | 1994 | | 2018 | | 1994 | | 2018 | | 1994 | | 2018 | |
| | GSV, m³/ha | Carbon Stock, t/ha | GSV, m³/ha | Carbon Stock, t/ha | GSV, m³/ha | Carbon Stock, t/ha | GSV, m³/ha | Carbon Stock, t/ha | GSV, m³/ha | Carbon Stock, t/ha | GSV, m³/ha | Carbon Stock, t/ha |
| *Pinus silvestris* | 271.0 | 117.9 | 1020.0 | 443.7 | 6744.0 | 2373.9 | 11,330.0 | 3988.2 | 1403.8 | 502.6 | 7090.0 | 2538.2 |
| *Picea abies* | 210.0 | 128.9 | 450.0 | 166.1 | - | - | - | - | - | - | - | - |
| *Larix sibirica* | 86.0 | 33.7 | 600.0 | 222.6 | 825.0 | 222.6 | 1230.0 | 306.1 | - | - | - | 0.0 |
| *Quercus robur* | - | - | - | | 911.0 | 447.3 | 1500.0 | 736.5 | 244.0 | 137.4 | 340.0 | 191.4 |
| *Ulmus sp.* | 103.0 | - | 190.0 | 118.6 | - | - | - | - | - | - | - | - |
| *Tilia cordata* | 90.0 | 56.2 | 140.0 | 87.4 | 670.0 | 225.1 | 1180.0 | 396.5 | 209.0 | 69.8 | 250.0 | 83.5 |
| *Betula pendula* | 452.0 | 197.5 | 660.0 | 288.4 | 1870.0 | 740.5 | 2950.0 | 1168.2 | 7254.0 | 2662.2 | 8240.0 | 3024.1 |
| *Populus temula* | 211.0 | 75.1 | 110.0 | 39.2 | 157.0 | 57.0 | 200.0 | 72.6 | 800.0 | 268.0 | 980.0 | 328.3 |

\* Coniferous and hardwood—0–40 years; softwood—0–20 years. \*\* Coniferous and hardwood—40–60 years; softwood—20–30 years. \*\*\* Coniferous and hardwood—60–100 years; softwood—30–80 years.

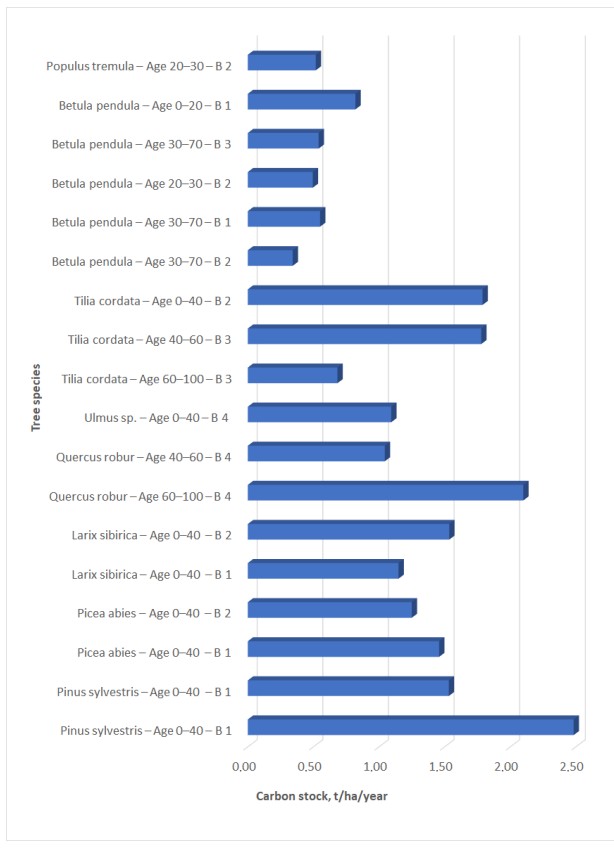

**Figure 8.** Average annual change in carbon stock (t/ha) in the forest species from 1994 to 2018 (B is bonitet).

Figures 9 and 10 show the spatial distribution of carbon stocks in the forests of Kandry-Kul Nature Park according to the materials of forest surveys of 1994 and 2018. The spatial patterns of carbon stocks correlated well with the GSV maps (Figure 7) and were closely related to the age of tree species.

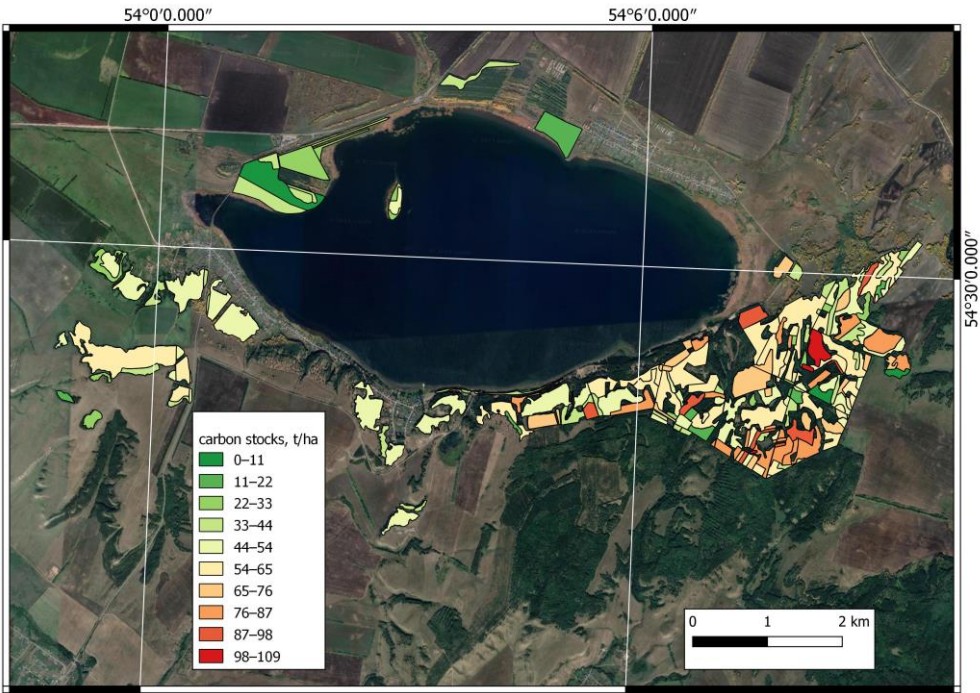

**Figure 9.** Spatial distribution of carbon stocks (t/ha) in the forests of Kandry-Kul Nature Park according to the materials of forest surveys of 1994.

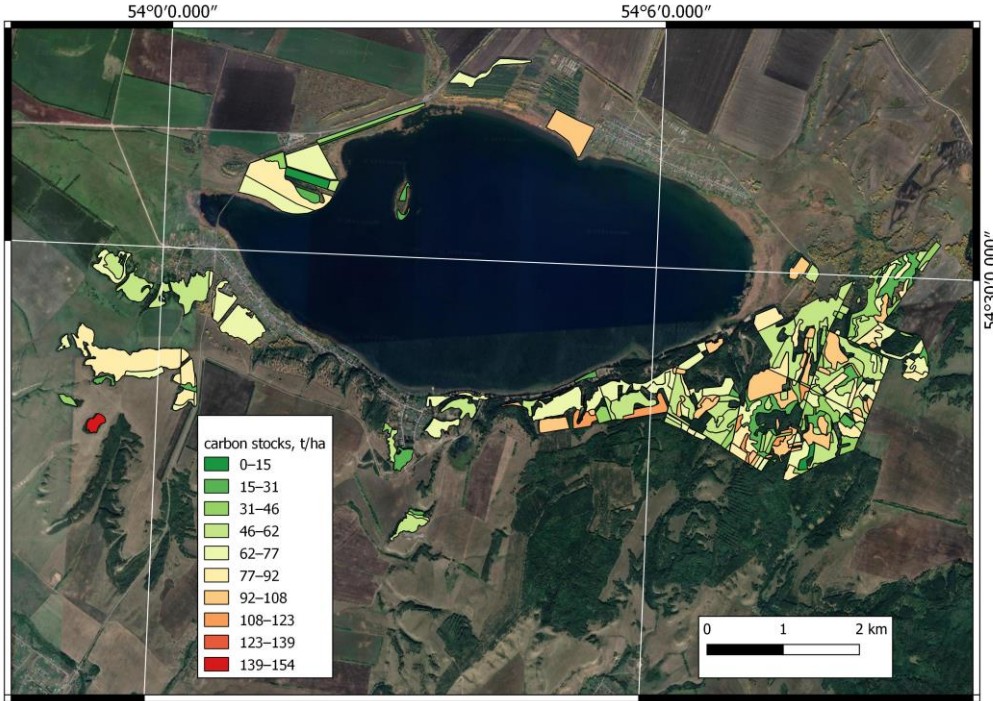

**Figure 10.** Spatial distribution of carbon stocks (t/ha) in the forests of Kandry-Kul Nature Park according to the materials of forest surveys of 2018.

### 4. Discussion

In our study, birch and pine were the most widely distributed tree species and contribute significantly to carbon storage. For example, Uri [22] reported that half of the total carbon stocks in a mature stand of silver birch (*Betula pendula*) were stored in the aboveground part of the trees in Southeastern Estonia. It has also been demonstrated that the carbon pool in silver birch biomass increased with stand age. In the American study of eastern white pines (Pinaceae: *Pinus strobus*), the authors concluded that after 80 years, the forest stores aboveground carbon in live trees at a high rate and continues to store large amounts of carbon in live trees for over 150 years [23].

Our results showed that the largest annual GSV was found in the taxation plots with young and middle-aged stands. Such forests grow faster and carbon accumulation in their biomass was more intensive compared with older stands. Moreover, stand density is important because it affects the rate and amount of GSV. For instance, Lin [24] reported that a non-thinned stand had a higher carbon stock than a medium or heavily thinned treatment plots of the same age in southern Taiwan. Similarly, it has been shown that a root carbon stock showed a significant positive relationship with stand density and basal area [25].

We found the negative annual GSV in some forest-taxation divisions, where birches were the predominant species. During the field observations in 2022 and analysis of remote sensing data, we found that in these habitats there was a bacterial disease caused by a phytopathogenic bacterium *Erwinia multivora* (Figure 10). This bacterial disease occurs in many parts of the world, including many regions of Russia [26–29]. Such factor significantly reduces the carbon stocks and, consequently, the annual GSV and carbon stock of the birch forests. We found that the main hotspots of the bacterium appeared after the hot and dry summer of 2010 and the following years of 2011 and 2012. Numerous works relate the distribution of this disease to changes in climatic conditions [28,30]. Moreover, among the above factors influencing the spread of the bacterium, soil conditions are also highlighted [27].

We presented a spatial-temporal GSV and carbon stocks mapping approach based on field-based observations according to taxation plots. Nevertheless, high-performance computers, software, and remote sensing data have significantly opened up opportunities for digital mapping of various ecosystem components. Remote sensing data is increasingly used in the assessment and mapping of GSV in many areas of the world. For example, Santoro et al. [21] presented an approach for estimating of GSV of the northern hemisphere (north of 10° N) using approximately 70,000 images from radar sensor on-board the Envisat satellite. Similarly, Zharko [31] used the red and near-infrared (NIR) bands of Sentinel-2 for a GSV estimation in the north-eastern part of Russian Kostroma region. However, the method of integrating field-based observations and remote sensing data to compare the results is considered to be the most accurate for GSV assessment [30–32]. An integration approach based on machine learning and multitemporal Landsat images was employed to dynamically estimate the spatial distribution of mangroves [33], where an increase in the area of trees over the 29th period was demonstrated. Tran et al. [13] applied to MODIS time-series remote sensing data to quantify the trend and rate of change to forest cover in the Central Highlands, Vietnam from 2001 to 2019. The authors found that deforestation significantly reduced tree cover at a rate of 0.76% per year, resulting in an overall loss of 14.5% in percent tree cover. Thus, especially valuable are archival data that allow us to compare and identify trends in the development of any processes.

At the same time, there are several limitations that can exist in studies evaluating and digitally mapping GSV and carbon stocks. Firstly, the accuracy and reliability of the digital mapping process depend on the resolution of the data used. If the spatial resolution of the data is low, it may not capture fine-scale variations in GSV and carbon stocks. Similarly, if the temporal resolution is too broad, it may not capture short-term changes or seasonal variations accurately. For instance, the time gap between the field observations in our study (1994 and 2018) may not capture all the changes that occurred in the forest during that period. Natural (disease, hurricane) or human-induced disturbances, such as

wildfires, logging, or afforestation, may have influenced the forest structure and composition, potentially affecting the accuracy of the GSV and carbon stock estimates [13,29,34]. Additionally, it is necessary to follow the same research methods for the correct interpretation and juxtaposition of archival and current data.

We demonstrated that Pinus systems contained a significantly higher total organic matter content in the ecosystem compared to Betula systems (592 and 421 mg/ha, respectively). At the same time, birch trees have been shown to retain nutrients better in the soil, making this species more promising for landscaping and carbon sequestration in Iceland [35]. The Betula and Pinus species are widely used to store carbon on degraded lands, such as in the area of metal mining [36,37]. According to Shanin et al. [38], the highest carbon uptake efficiency for mixed stands is predicted for common pine with 20–30% of small-leaved species (*Betula spp.* and *Populus tremula*). At the same time, earlier studies [39] stated that the growth of net carbon production in mixed plantations was higher and increased with an increase in the share of pine in Finland. Thus, our studies also confirm the greater carbon storage potential of pine stands.

## 5. Conclusions

In this study, we presented the results of GSV and carbon stocks assessment and spatio-temporal digital mapping on the example of the Kandry-Kul Natural Park (Republic of Bashkortostan, Russia). We identified the predominant tree species by forest taxation and evaluated the GSV and carbon stocks using 1994 and 2018 field observations. We showed that the predominant species were birch and pine, which contributed most to carbon storage. Based on visualization of spatial patterns, we revealed that the maximum GSV annual growth between 1994 and 2018 was 6.4–7.9 $m^3$/ha and occurred in areas with a fairly small GSV in 1994. These forest taxation areas were characterized by young stands of predominantly pine trees, with an annual growth of GSV 4.2 $m^3$/ha per year. The change in carbon stocks for pine stands was on average 2 t/ha per year. Birch forest were characterized by mature stands with lower annual growth of GSV (1.2 $m^3$/ha) and provided a secondary contribution to the potential of carbon storage. However, there were bacterial diseases, negatively affecting the GSV and annual growth, as well as on the change in the carbon stocks. A large number of dead birch forest affected by bacterial diseases explain this sharp decline in GSV values. For birch stands, the change in carbon stock was approximately 0.5 t/ha per year, which is 30% of the average values for the forest-steppe region Thus, we conclude that the next stage of verification of state forest inventory data requires field or remote assessments.

Archival data are a valuable source for scientific research and spatio-temporal mapping, which allows us to compare and identify trends in the development of environmental processes. Assessment of the spatial distribution and mapping of GSV is essential to climate change mitigation and sustainable land management. This study emphasizes that climate change is causing significant disruptions to natural systems, including forests, and studying forest GSV and carbon stocks can help us understand and address the impacts of these changes on ecosystems and their resilience to future climatic conditions.

**Author Contributions:** Conceptualization, A.V. and L.B.; methodology, A.V., A.S. and E.B.; software, A.V., A.S. and E.B.; validation, I.T. and R.S.; formal analysis, I.T. and R.S.; investigation, A.V. and L.B.; resources, L.B. and R.S.; data curation, A.V., L.B. and I.T.; writing—original draft preparation, A.V., A.S. and D.M.; writing—review and editing, E.B.; visualization, A.S., E.B. and D.M.; supervision, A.V., L.B. and I.T.; project administration, L.B., I.T. and R.S.; funding acquisition, L.B. All authors have read and agreed to the published version of the manuscript.

**Funding:** This study was funded by the Ministry of Science and Higher Education of the Russian Federation "PRIORITY 2030" (National Project "Science and University").

**Data Availability Statement:** Not applicable.

**Conflicts of Interest:** The authors declare no conflict of interest.

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
