# Peer review of "Spatio-Temporal Analysis of Forest Growing Stock Volume and Carbon Stocks: A Case Study of Kandry-Kul Natural Park, Russia"

_land, doi:10.3390/land12071441_

Round 1
Reviewer 1 Report
Dear authors.
I am grateful to review the article "Spatial-temporal Analysis of Forest Growing Stock Volume and Carbon Stocks: A Case Study of "Kandry-Kul" Natural Park, Russia". I think that the article is good, it is devoted to an actual topic.
I will note some comments that will improve the quality of the article.
1. Why do you use "" in the name of Natural Park?
2. In section 1, very little attention is paid to the history of the issue. It is necessary to significantly strengthen the literature review.
3. Design the drawings in the same style. Specify coordinate grids on all of them. Indicate the direction to the north.
4. Why in Figure 5 do you use a classification step through 34 m3/ha?
5. Figure 9 should be divided into 2 separate figures. They merge and are poorly perceived.
6. In section 4, it is necessary to indicate the limitations of the study, to make comparisons with other regions of the world, to indicate the prospects for further research. I.e., to conduct a more in-depth analysis.
Author Response
Dear Reviewer 1. We thank you for reviewing our work and for your valuable comments. Below are the answers to your questions.
Point 1
Why do you use "" in the name of Natural Park?
Response 1
Thank you very much for the comment. We have made the appropriate corrections
Point 2
In section 1, very little attention is paid to the history of the issue. It is necessary to significantly strengthen the literature review.
Response 2
Thanks for the suggestion. We improve the introduction part with emphasis on the literature review. Please see lines 54-67.
Point 3
Design the drawings in the same style. Specify coordinate grids on all of them. Indicate the direction to the north.
Response 3
Thank you for your comment. We have added a north arrow and a coordinate grid to the all maps.
Point 4
Why in Figure 5 do you use a classification step through 34 m3/ha?
Response 4
Thank you for your question. We understand the need for clarification. For clarity, the spatial distribution of GSV is divided into 10 classes, for 1994 the step was 25 m/ha, and for 2018 the step was 34 m/ha. This information is presented in the text of the article - lines 189-190.
Point 5
Figure 9 should be divided into 2 separate figures. They merge and are poorly perceived?
Response 5
Thank you for the comment. Figure 9 is now divided into Figure 9 and 10.
Point 6
In section 4, it is necessary to indicate the limitations of the study, to make comparisons with other regions of the world, to indicate the prospects for further research. I.e., to conduct a more in-depth analysis
Response 6
Thank you for the suggestion. We improved this part and added the comparisons and limitations of such approaches. Please see lines 327-334 and 336-347.

Reviewer 2 Report
Studying Spatio-temporal patterns of Forest Growing Stock Volume and Carbon Stocks in Russian forest is an important topic because few studies focus on forest carbon stock in global carbon stock database. The authors conducted forest inventory in both 1994 and 2018 and calculated the spatiotemporal patters of volume and carbon stock, which could be important understand the dynamic change of forest carbon stock under both climate change and human activities. However, I have several concerns before it can be accepted for publication.
1. The method section need further clarification: More information on method on plot design, starting diameter of measured tree; how about small tree or understory?
2. “radar remote sensing data Alos-Palsar [15]. These satellite raster images represent the value of GSV (m3/ha) and biomass (t/ha).” I can not understand. Because you measured all trees, and the biomass could directly be calculated from measurement. Therefore, more details how to calculate biomass and volume, or how to convert volume to biomass and carbon stock should be given. Was biomass from satellite raster images validated by field observations? How about the data quality? Why did you use radar remote sensing data?
3. The language need further improvement and sometimes it is difficult to understand.
Minor issues:
L137: Please show the conversion factors;
The north arrow or longitude or latitude are missing for Fig. 2, 3, 5, 7 and 9
I cannot clearly understand Fig. 4 and 6. Does the frequency from your plot data? Or Radar data? Or polygon data?
It is unclear whether your data analysis was conducted based on field inventory or Radar data. Please clarify in method section.
Fig. 10 does not make sense, and can move it to supplementary
Data Availability: it would be great helpful to share the field observational data in global database because there is always a lack of field data from Russia.
The language need further improvement and sometimes it is difficult to understand.
Author Response
Dear Reviewer 2. We thank you for reviewing our work and for your valuable comments. Below are the answers to your questions.
Point 1
The method section need further clarification: More information on method on plot design, starting diameter of measured tree; how about small tree or understory
Response 1
Thanks for the comment. We agree that the methods are not sufficiently disclosed. According to the forest taxation methodology, trees less than 5 m in height and with a trunk diameter of less than 5 cm were not included in the study. The information was added to the text - lines 122-124
Point 2
“radar remote sensing data Alos-Palsar [15]. These satellite raster images represent the value of GSV (m3/ha) and biomass (t/ha).” I can not understand. Because you measured all trees, and the biomass could directly be calculated from measurement. Therefore, more details how to calculate biomass and volume, or how to convert volume to biomass and carbon stock should be given. Was biomass from satellite raster images validated by field observations? How about the data quality? Why did you use radar remote sensing data?
Response 2
Thanks. We agree with the comment. The radar survey data are not considered within the scope of this article and are not consistent with its context. This sentence is deleted from the text. At the same time, we understand the importance of using remote sensing methods and analyzing radar survey data, and we will try to present the results of measurements in other publications.
Point 3
The language need further improvement and sometimes it is difficult to understand.
Response 3
Thank you for the comment, we have reviewed and edited the text according to the English language.
Point 4
L137: Please show the conversion factors
Response 4
Thanks for the comment, the necessary information has been added to Table 1.
Point 5
The north arrow or longitude or latitude are missing for Fig. 2, 3, 5, 7 and 9
Response 5
Thank you for the comment. Changes have been made accordingly.
Point 6
I cannot clearly understand Fig. 4 and 6. Does the frequency from your plot data? Or Radar data? Or polygon data?
Response 6
Thank you for your comment. As stated above, in this article we present the results of ground surveys only. Apparently due to a technical error, a sentence using radar data was included in the text of the article. At the same time, we understand the importance of using radar data analysis and are currently working to verify remote data with the results of field studies. In the future, we plan to prepare relevant publications.
Point 7
It is unclear whether your data analysis was conducted based on field inventory or Radar data. Please clarify in method section.
Response 7
Thank you very much for your note. Our article contains the results of field surveys, which have already been further analyzed in the geographic information system.
Point 8
Fig. 10 does not make sense, and can move it to supplementary
Response 8
Thanks more for the comment. We have removed this fig. from the article.
Point 9
Data Availability: it would be great helpful to share the field observational data in global database because there is always a lack of field data from Russia
Response 9
Thank you for this suggestion. We will try to add the results of our research to the free access.

Round 2
Reviewer 1 Report
Accept in present form
Author Response
Dear Reviewer 1. We thank you for your positive assessment with our work and for your valuable comments.
Reviewer 2 Report
Table 1: I suggest: Conversion factors for calculating carbon stocks in tons/ha
Again regarding to Figures only "north arrow" or "longitude/latitude", not both.
Again, I wish the author share the dataset because there is a lack of Russian forest.
These isusses are small and I do not need to read the revised manuscript again.
Author Response
Dear Reviewer 2. We thank you for your positive assessment with our work and for your valuable comments. Below are the answers to your questions.
Point 1
Table 1: I suggest: Conversion factors for calculating carbon stocks in tons/ha
Response 1
Thanks for the comment. We improved the name of table 1 as your suggestion.
Point 2
Again regarding to Figures only "north arrow" or "longitude/latitude", not both
Response 2
Thanks more for the comment. We deleted «north arrow» from figures.
Point 3
Again, I wish the author share the dataset because there is a lack of Russian forest.
Response 3
Thank you for more this suggestion. Of course, we will try to share the dataset.